# Linking glycemic dysregulation in diabetes to symptoms, comorbidities, and genetics through EHR data mining

Isa Kristina Kirk[1†], Christian Simon[1†], Karina Banasik[1], Peter Christoffer Holm[1], Amalie Dahl Haue[1], Peter Bjødstrup Jensen[1,2], Lars Juhl Jensen[1], Cristina Leal Rodríguez[1], Mette Krogh Pedersen[1], Robert Eriksson[1], Henrik Ullits Andersen[3], Thomas Almdal[3,4], Jette Bork-Jensen[5], Niels Grarup[5], Knut Borch-Johnsen[6], Oluf Pedersen[3,5], Flemming Pociot[3,7], Torben Hansen[3,5], Regine Bergholdt[3], Peter Rossing[3,8]*, Søren Brunak[1,9]*

[1]Novo Nordisk Foundation Center for Protein Research, University of Copenhagen, Copenhagen, Denmark; [2]Odense Patient Data Explorative Network (OPEN), Odense University Hospital, Odense, Denmark; [3]Steno Diabetes Center Copenhagen, Gentofte, Denmark; [4]Department of Endocrinology, Rigshospitalet, Copenhagen, Denmark; [5]Novo Nordisk Foundation Center for Basic Metabolic Research, University of Copenhagen, Copenhagen, Denmark; [6]Holbæk Hospital, Holbæk, Denmark; [7]Department of Clinical Medicine, Herlev-Gentofte Hospital, Herlev, Denmark; [8]Department of Clinical Medicine, University of Copenhagen, Copenhagen, Denmark; [9]Center for Biological Sequence Analysis, Department of Bio and Health Informatics, Technical University of Denmark, Lyngby, Denmark

*For correspondence:
peter.rossing@regionh.dk (PR);
soren.brunak@cpr.ku.dk (SB)

†These authors contributed
equally to this work

Competing interests: The
authors declare that no
competing interests exist.

Reviewing editor: Alfonso
Valencia, Barcelona
Supercomputing Center - BSC,
Spain

**Abstract** Diabetes is a diverse and complex disease, with considerable variation in phenotypic manifestation and severity. This variation hampers the study of etiological differences and reduces the statistical power of analyses of associations to genetics, treatment outcomes, and complications. We address these issues through deep, fine-grained phenotypic stratification of a diabetes cohort. Text mining the electronic health records of 14,017 patients, we matched two controlled vocabularies (ICD-10 and a custom vocabulary developed at the clinical center Steno Diabetes Center Copenhagen) to clinical narratives spanning a 19 year period. The two matched vocabularies comprise over 20,000 medical terms describing symptoms, other diagnoses, and lifestyle factors. The cohort is genetically homogeneous (Caucasian diabetes patients from Denmark) so the resulting stratification is not driven by ethnic differences, but rather by inherently dissimilar progression patterns and lifestyle related risk factors. Using unsupervised Markov clustering, we defined 71 clusters of at least 50 individuals within the diabetes spectrum. The clusters display both distinct and shared longitudinal glycemic dysregulation patterns, temporal co-occurrences of comorbidities, and associations to single nucleotide polymorphisms in or near genes relevant for diabetes comorbidities.

## Introduction

Electronic Health Records (EHRs) contain patient characteristics from different data layers including text narratives, assigned diagnosis codes, biochemical values, and prescription data. These data types display a high degree of complementarity, providing an excellent basis for deep phenotyping and patient stratification. Recent studies have shown how structured data derived from EHRs can be used to assess phenotypic variability of different disease areas (*Li et al., 2015*; *Dahlem et al., 2015*;

*Doshi-Velez et al., 2014*; *Kho et al., 2011*; *Kho et al., 2012*). While the use of structured EHR data in many instances resembles traditional registry- or biobank-based research, the inclusion of unstructured data such as clinical narratives allows for the definition of even more fine-grained phenotypes, which could lead to novel subgroup stratifications (*Li et al., 2015*; *Roque et al., 2011*; *Miotto et al., 2016*).

A vast amount of information on symptoms, lifestyle, complications, and comorbidities is available from clinical narratives in unstructured EHR data. Text mining applying natural language processing (NLP) algorithms is one strategy, but simpler approaches have also been shown to be valuable in the context of clinical text, for reviews see *Jensen et al. (2012)* and *Denny (2012)*. These methods work across language barriers and have been successfully implemented in for example adverse drug reaction detection (*Warrer et al., 2012*), subgrouping of chronic obstructive pulmonary disease (*Fu et al., 2015*), cancer subgrouping (*Chen et al., 2015*), and classification of epileptic children (*Pereira et al., 2013*). Such studies show the possibilities of using and integrating different parts of EHRs for matching phenotypically similar subgroups to biomarker data, which is key to improved treatment and characterizing etiological differences.

Several large initiatives have been established for utilizing EHRs, including the Electronic Medical Records and Genomics (eMERGE) consortium of DNA biorepositories that links genetic data with electronic medical records (*McCarty et al., 2011*; *Gottesman et al., 2013*), and EMR-driven non-negative restricted Boltzmann machines (eNRBM) which use unsupervised learning for analyzing EHRs (*Tran et al., 2015*). Furthermore, other studies have used general approaches for finding direct and inverse comorbidities (*Doshi-Velez et al., 2014*; *Roque et al., 2011*; *Gligorijevic et al., 2016*).

Diabetes Mellitus (DM) is a difficult disease to stratify (*American Diabetes Association, 2017*). DM covers etiologically different metabolic disorders that exhibit the same phenotype, hyperglycemia, due to either insufficient insulin production relative to insulin demand or insulin resistance. Although DM is classified into different major subtypes, it has been hypothesized to represent a disease continuum rather than strict distinct disease subtypes (*American Diabetes Association, 2017*; *Flannick et al., 2016*). One recent data-driven study used five subgroups of adult-onset diabetes and clustered six parameters from the structured data of the EHR (*Ahlqvist et al., 2018*). DM is a complex disorder associated with several comorbidities and organ complications. These can be classified as macrovascular complications that is cardiovascular disease, and microvascular complications resulting in eye, kidney, and nerve damage. Cardiovascular complications alone are responsible for 50–80% of all-cause mortality in diabetes patients (*Laakso, 2001*). The severity of complications is affected by glycemic dysregulation, that is increased or fluctuating blood glucose levels (*Stratton et al., 2000*; *UK Prospective Diabetes Study Group, 1998a*; *UK Prospective Diabetes Study Group, 1998b*; *Nathan et al., 1993*), and successful reduction and prevention of diabetic complications have been observed when the glycemic dysregulation is reduced or removed (*Stratton et al., 2000*; *UK Prospective Diabetes Study Group, 1998a*). Therefore, risk factors for glycemic dysregulation are crucial to diabetes progression (*Ahlqvist et al., 2015*). Known risk factors for complications include age, diabetes duration, polypharmacy, comorbidities (*Juarez et al., 2012*), increased levels of circulating triglyceride and LDL-cholesterol, and lower levels of HDL-cholesterol (*Saudek et al., 2006*; *Giannini et al., 2011*; *Bitzur et al., 2009*). Finding new risk factors that can help classify poorly regulated versus well-regulated diabetes, such as other biochemical variables or genetic variants, could improve treatment and reduce diabetic complications.

In this study, we utilized the unstructured data of EHRs and performed a deep phenotypic characterization of a Danish diabetes cohort of 14,017 individuals, aged 18 to 101 at the end of the study, using vocabularies comprising both diagnosis codes and 'exposome' related terms. We used text-mined and assigned diagnosis codes to stratify the cohort and described it using both physiological and genetic variation data. The unstructured EHR data enabled us to classify patients based on their level of glycemic dysregulation and to identify potential biochemical and genetic markers associated with dysglycemia.

## Results

### Text mining the EHR corpus

The general aim of the text mining effort was to obtain a richer phenotypic characterization of each patient. Initially, each patient had in 4.9 assigned codes on average. Applying text mining with two vocabularies (ICD-10 and SDC-custom) resulted in a 4-fold increase to 18.6 codes per patient. Moreover, the distribution of codes across ICD-10 chapters changed considerably when adding the text-mined codes, with chapters I, VII, XVIII and XIX showing the largest increases (6, 15, 25 and 22-fold increase, respectively) (*Figure 1*). This illustrates the difference between the assigned diagnosis codes from the structured data and the much more symptom-rich codes detected by text mining.

### Comorbidity clustering based on text-mined and assigned diagnosis codes

For each patient, the assigned and text-mined ICD-10 codes were combined to create a patient-specific diagnosis-vector where the primary diabetes type (E10 or E11) was not included. Contrary to cancer for example, where the ICD-10 diagnoses are quite reliable and highly detailed, the primary codes in a multi-organ disease like diabetes are used in a fuzzier way, as the knowledge on robust

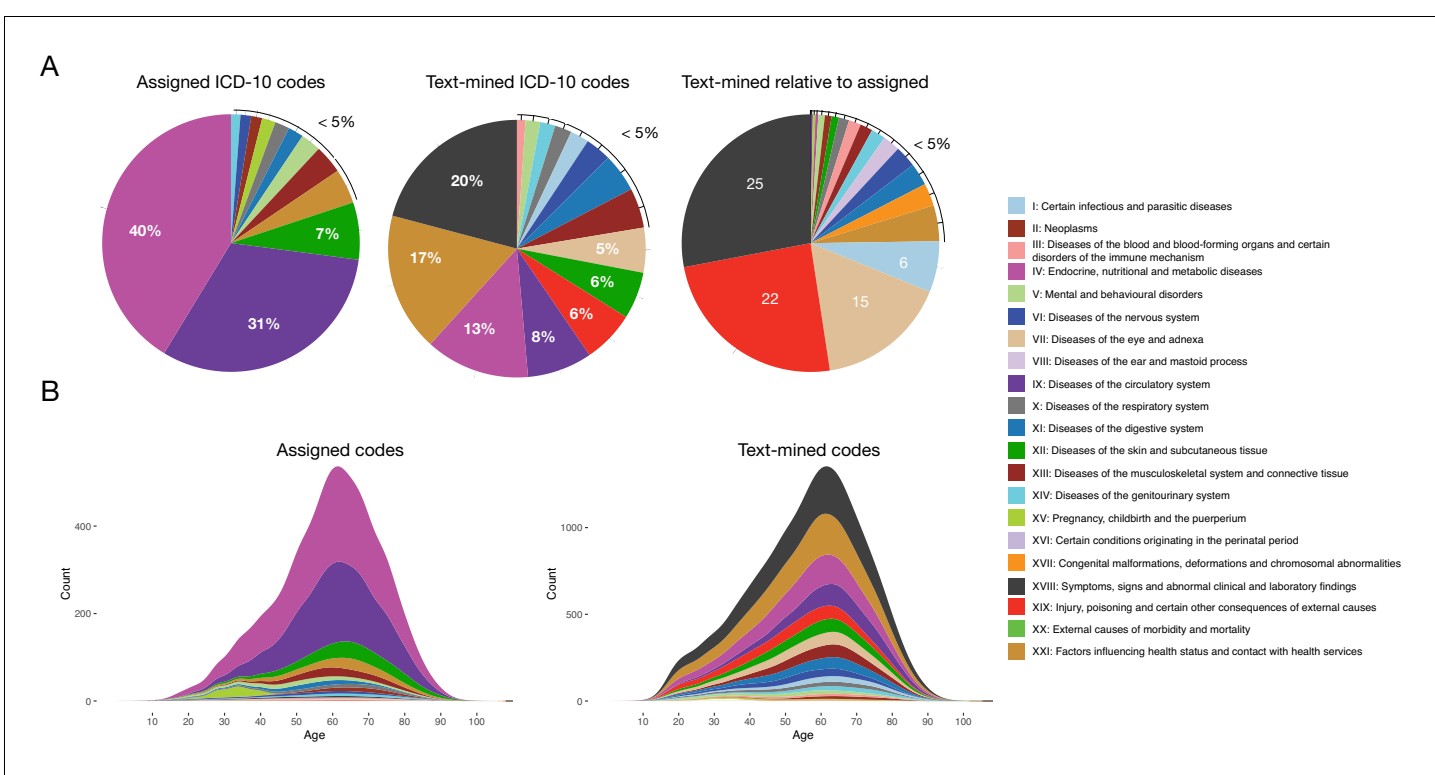

**Figure 1.** Comparison of distributions of ICD-10 diagnosis codes with and without text mining. (**A**) Percentage of diagnosis codes belonging to the different ICD-10 chapters and the relative increase in diagnosis codes from the different chapters when combining the text-mined and assigned codes. (**B**) Age distributions of text-mined and assigned ICD-10 diagnosis codes from the SDCC corpus divided into the 21 ICD-10 chapters. The online version of this article includes the following source data and figure supplement(s) for figure 1:

**Source data 1.** Diagnosis code breakdown data.

**Source data 2.** Age distribution data.

**Figure supplement 1.** Distribution of patients per physiological and biochemical test.

**Figure supplement 2.** Physiological and biochemical tests in the SDCC corpus.

**Figure supplement 3.** Linear Discriminant Analysis 1 (LDA).

**Figure supplement 4.** Linear Discriminant Analysis 2 (LDA).

**Figure supplement 5.** Distribution of HbA1c measurements for T1D and T2D patients.

**Figure supplement 6.** Biochemical patterns for the level of glycemic dysregulation.

diabetes subtypes and their characteristics in the context of comorbidities is quite limited. We do therefore not want the clustering to be driven by the broad, less etiology-relevant primary codes from the endocrinology chapter, but rather by more objectively observed symptoms, other diseases and lifestyle features. Following code-abundance normalization and BM-25 correction the vectors were clustered using MCL producing 172 clusters (mean = 65 patients, min = 11, max = 979, median = 40), in which 11,208 patients (80.47%) were included *Figure 2A*. The remaining 2720 patients (19.53%) were in clusters with ten or less patients and were therefore omitted from subsequent analyses.

Even though codes for the primary diabetes type were not part of the diagnosis vectors, specific clusters were significantly enriched for T1D patients (cluster 1: N = 506, adj. p-value=9.3e-51 and cluster 9: N = 101 adj. p-value=1.2e-10). Other clusters had significantly more T2D patients than expected (cluster 3: N = 233, adj. p-value=9.1e-10, cluster 5: N = 170, adj. p-value=3.8e-13 and cluster 6, N = 158 adj. p-value=8.4e-17). In addition, we observed a cluster significantly enriched with the ICD-10 term E13: *other diabetes* (cluster 25, N = 93, adj. p-value=1.8e-142), which includes diabetes due to genetic defects, post-pancreatectomy diabetes and post-procedural diabetes. Several other clusters had a mix of T1D and T2D patients according to the assigned codes. Further characteristics of the laboratory data and prescription data as well as the clusters regarding sex, age, observational time, years with diabetes etc. can be found in *Supplementary files 1–3* and in *Figure 1—figure supplement 1*, *Figure 1—figure supplement 2*, *Figure 1—figure supplement 3*, *Figure 1—figure supplement 4*, *Figure 2—figure supplement 1*, *Figure 2—figure supplement 2*, *Figure 2—figure supplement 3*, *Figure 2—figure supplement 4*. The robustness of the clustering was found to be high (see description in Materials and methods and *Figure 2—figure supplement 5*). To maintain

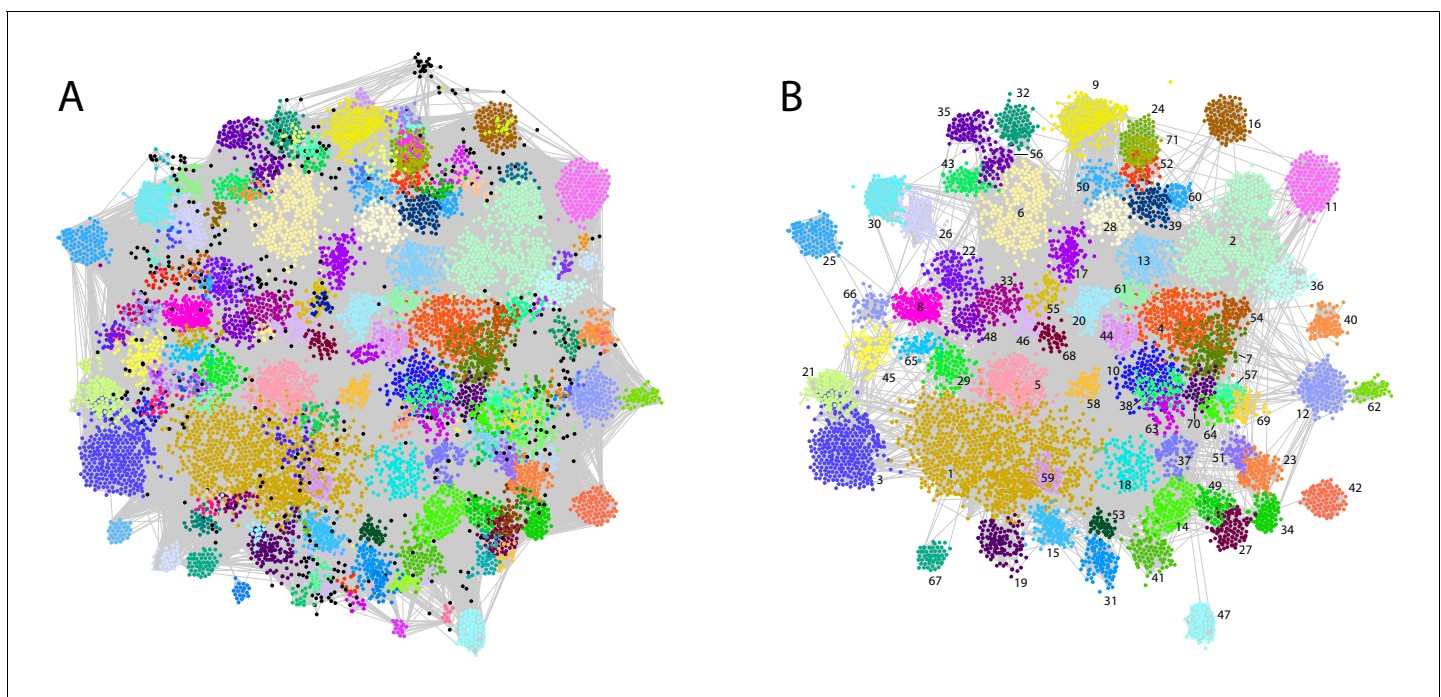

**Figure 2.** Phenotypic clusters found in the SDCC cohort. The clustering was created with diagnosis vectors of 13,928 patients (with text in the record) comprising both text-mined and assigned ICD-10 codes. A total of 172 clusters were created, where 11,208 patients (80.47%) were captured in the clustering (clusters with five or less patients were discarded for statistical reasons). (**A**) Each node represents a patient within the corpus colored by the association to one of the 172 unique clusters. (**B**) The 71 clusters with at least 50 patients colored with the same palette as in (**A**).

The online version of this article includes the following figure supplement(s) for figure 2:

**Figure supplement 1.** Density of days in contact with SDCC for each cluster.

**Figure supplement 2.** Distribution of assigned primary diabetes type for each cluster.

**Figure supplement 3.** Distribution of age for each cluster.

**Figure supplement 4.** Distribution of duration of diabetes for each cluster.

**Figure supplement 5.** Clustering robustness analysis.

power in subsequent analyses we focused on clusters with at least 50 patients (71 clusters comprising 8652 patients, *Figure 2B*).

## Enriched comorbidity and symptom patterns in diabetes patient clusters

The 71 clusters (*Figure 2B*) were grouped by hierarchical clustering, using distances obtained from cluster specific symptoms from the ICD-10 chapter XVIII (level 1). Six main groups and an outlier (cluster 70) were found containing 5, 8, 21, 11, 7 and 18 of the original clusters, respectively. The symptom groups are illustrated by the branch colors in *Figure 3*. The nodes represent the 71 clusters each depicted as a pie chart displaying the comorbidities and symptoms that are significantly enriched (adj. p-value≤0.05), see *Supplementary file 4* for details on the enrichment and p-values.

The 71 clusters were defined based on the associated comorbidities, excluding DM without complications, and from the pie charts we observed that distinct diagnoses do indeed characterize the clusters. For example, ICD-10 code N40: *Benign prostatic hyperplasia* for cluster 56, L40: *Psoriasis* for cluster 16, F20: *Schizophrenia* for cluster 47, K29: *Functional intestinal disorders* for cluster 17, and Z94: *Transplanted organ and tissue status* for cluster 42. Using Fisher's exact test, we found that: *Symptoms related to skin and subcutaneous tissue* (adj. p-value<0.001) characterized symptom group five and *Symptoms related to digestive system and abdomen; cognition, perception, emotional state and behavior;* and *general symptoms and signs* (adj. p-value<0.001 for all) characterized symptom group 3. These results correspond well to the enriched codes observed in *Figure 3*, as was the case for the other enriched codes across the 71 clusters within the six symptom groups.

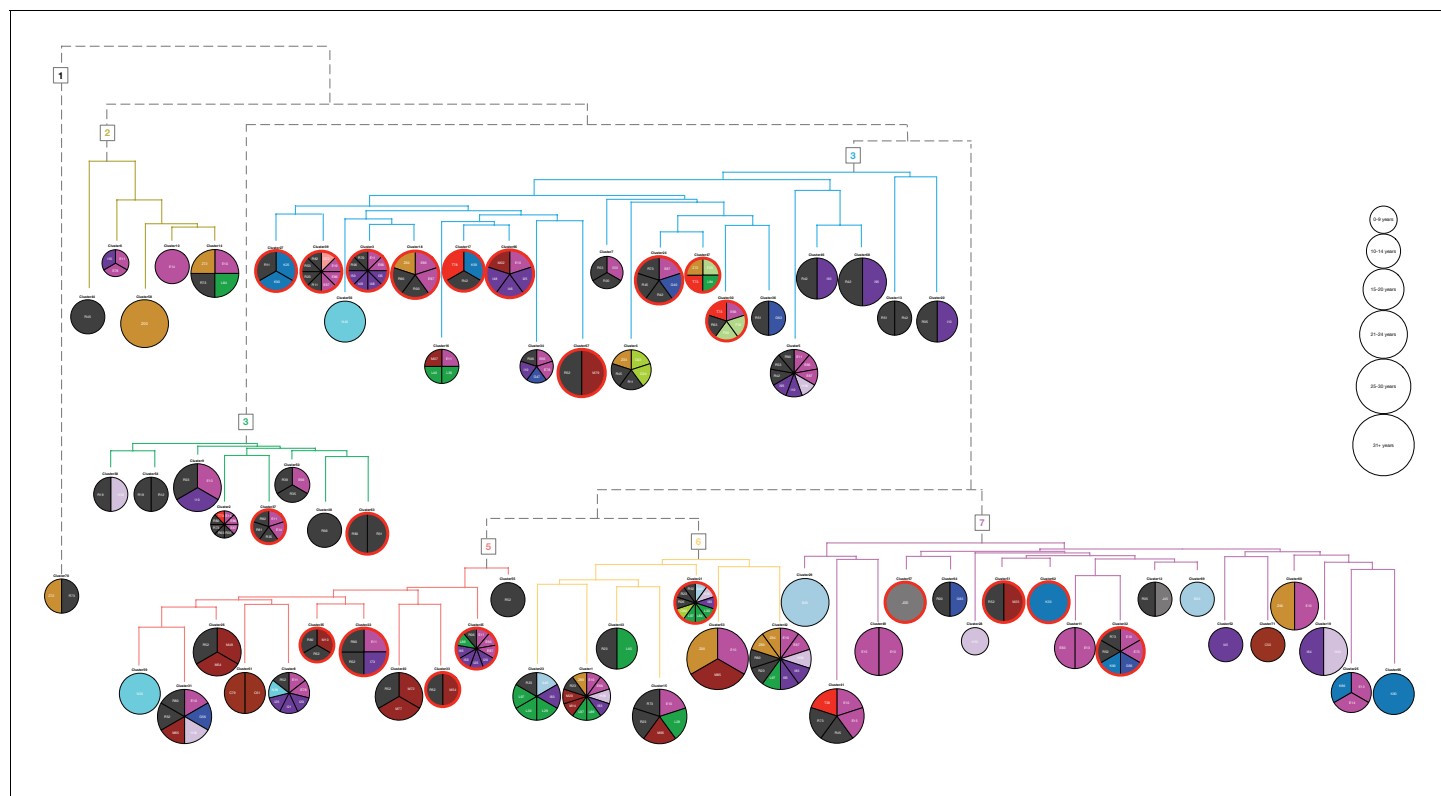

**Figure 3.** Hierarchical clustering based on enriched comorbid ICD-10 diagnoses. The comorbidities present in a minimum of 10 patients and significantly enriched (adj. p-value<=0.05) in each cluster are shown in the pie charts. The number of significant codes ranges from 1 to 10. Each color corresponds to an ICD-10 code chapter as listed in the legend of *Figure 1*. Six main groups and an outlier (cluster 70) resulted, and the colors of the dendrogram branches indicate to which hierarchical groups the clusters belong. The size of the pie charts represents the average diabetes duration (years with diabetes) divided into six bins. The 21 clusters where at least 50% of the patients have three or more HbA1c severity parameters are marked with a red line surrounding the pie chart.

## Genomic characterization by SNP association of phenotypically determined clusters

We evaluated the 71 clusters in the six symptom groups, plus the outlier cluster, for SNPs that could characterize the different groups (details on the genetic data can be found in the 'Genomic characterization' section under Materials and methods). The five highest association signals (independent) for each group are shown in *Supplementary file 5*. Only results from analyses with more than 15 cases and a well-calibrated QQ-plot (visual inspection and a lambda inflation factor >0.96) are reported. Accordingly, clusters 1–5, 7–9, 12, 15–18, 21–23, 26, 31, 35, 39, 45, 46, and 66, as well as all aggregated symptom groups, met the criteria. The median coverage of the symptom clusters was 31% [range: 10–67]. SNPs characterizing the symptom groups were found in several instances and association signals to disease-associated genes were also found for several of the clusters (*Figure 3*). Most frequently found, unsurprisingly, were genes associated by GWAS to diabetes or diabetes-related cardio-metabolic traits (cluster 3: *MYO3B*, cluster 4: *DAPK1*, cluster 5: *LPIN2*, cluster 7: *SAMD4A* and *FHIT*, cluster 8: *ERG* and *PLCB1*, cluster 12: *MYT1L*, cluster 15: *UBE2WP1*, cluster 16: *ADARB2*, *CDKAL1*, and *CLIP1*, cluster 17: *C8orf37-AS1*, cluster 21: *FHOD3* and *MCF2L*, cluster 24: *MTCL1*, cluster 26: *NTM*, cluster 31: *PCDH15*, *CDH4*, and *DCTD*, cluster 31: *KLF12*, cluster 39: *FHOD3*, cluster 45: *IGF1R*, *BCAS3,* and *TENM4*, cluster 46: *NRXN3*). Cluster eight is characterized by cardiovascular complications, and three of the top ranking genes for this cluster have been associated with LDL peak particle diameter (*THBS4*; *Rudkowska et al., 2015*), abdominal aortic aneurysm (*ERG*; *Jones et al., 2017*), pulse pressure (*ERG*; *Warren et al., 2017*), and diastolic blood pressure (*PLCB1*; *Warren et al., 2017*). Cluster 21 is enriched for the ICD-10 diagnosis foot ulcer (L97), and *MCF2L*, one of the top ranking genes for cluster, has been associated with both end-stage coagulation (*Williams et al., 2013*) and prothrombin time (*Tang et al., 2012*). In total, of the top five association signals that were mapped to genes (n = 103) we found five (*CDKAL1, DCDC2C, KLF12, LPIN2, TLE1*) to be related with diabetes.

## Comorbidity pairs and patterns within symptom related clusters

We detected codes occurring significantly more or less together within and across the symptom groups (Fischer's test with Bonferroni adjusted p-values<=0.01) defining distinct comorbidity pairs. If the comorbidity pairs covered more than 100 unique codes (symptom groups 4 and 7) we extracted only the most significant pairs until these pairs consisted of 100 unique codes.

*Figure 4A* illustrates the comorbidity correlations for the six main symptom groups where each pairwise interaction has a comorbidity score (see Material and methods). To characterize whether a diagnosis occurred significantly more before or after another, we made this analysis in a temporal manner. *Figure 4B* illustrates the comparison of the first diagnosis (row) to the second diagnosis (column). We found that especially the diagnoses related to diabetes (E13, O24), diabetes with complications (shortened to E10 and E11), obesity (E66), diseases of the pancreas (K86), poly- and proteinuria (R35 and R80), and to some extent hypertension and ischemic heart disease (I10, I20, I21, I25) are observed before other diagnoses (blue indicates that the row diagnosis is observed prior to the column diagnosis more than expected, and red indicates the opposite). Focusing on the different symptom groups, we detected which comorbidity pairs were unique in the different groups, and *Figure 4C* displays these unique comorbidity interactions.

In symptom group two we found that L84: *corns and callosities* is observed significantly more together within patients with T1D than T2D (CS = 1.24, adj. p-value=4.06e-15 and CS = −1.58, adj. p-value=1.25e-03, respectively). Temporal analysis of diagnosis occurrence showed that T1D is observed before L84 (*Figure 4B*, mean time difference = 8.3 years, adj. p-value=1.01e-39). Corns or callosities are unproblematic in healthy people, but in diabetes patients they can cause skin defects that increase the risk for additional complications, for example foot ulcers which can lead to amputations (*Apelqvist et al., 2000*; *Hunt, 2011*).

Although not observed significantly together within any clusters the temporal analysis showed that the time between T2D and elevated blood glucose levels (R73) is significantly shorter in symptom group two than in groups 4, 5 and 6 (mean time = 2.2 days; adj. p-value=6.45e-04, 3.29e-06 and 2.73e-06, respectively).

In symptom group 5, five of the eleven clusters are enriched with ICD-10 codes from chapter XIII: *Diseases of the musculoskeletal system and connective tissue*, especially dorsopathies,

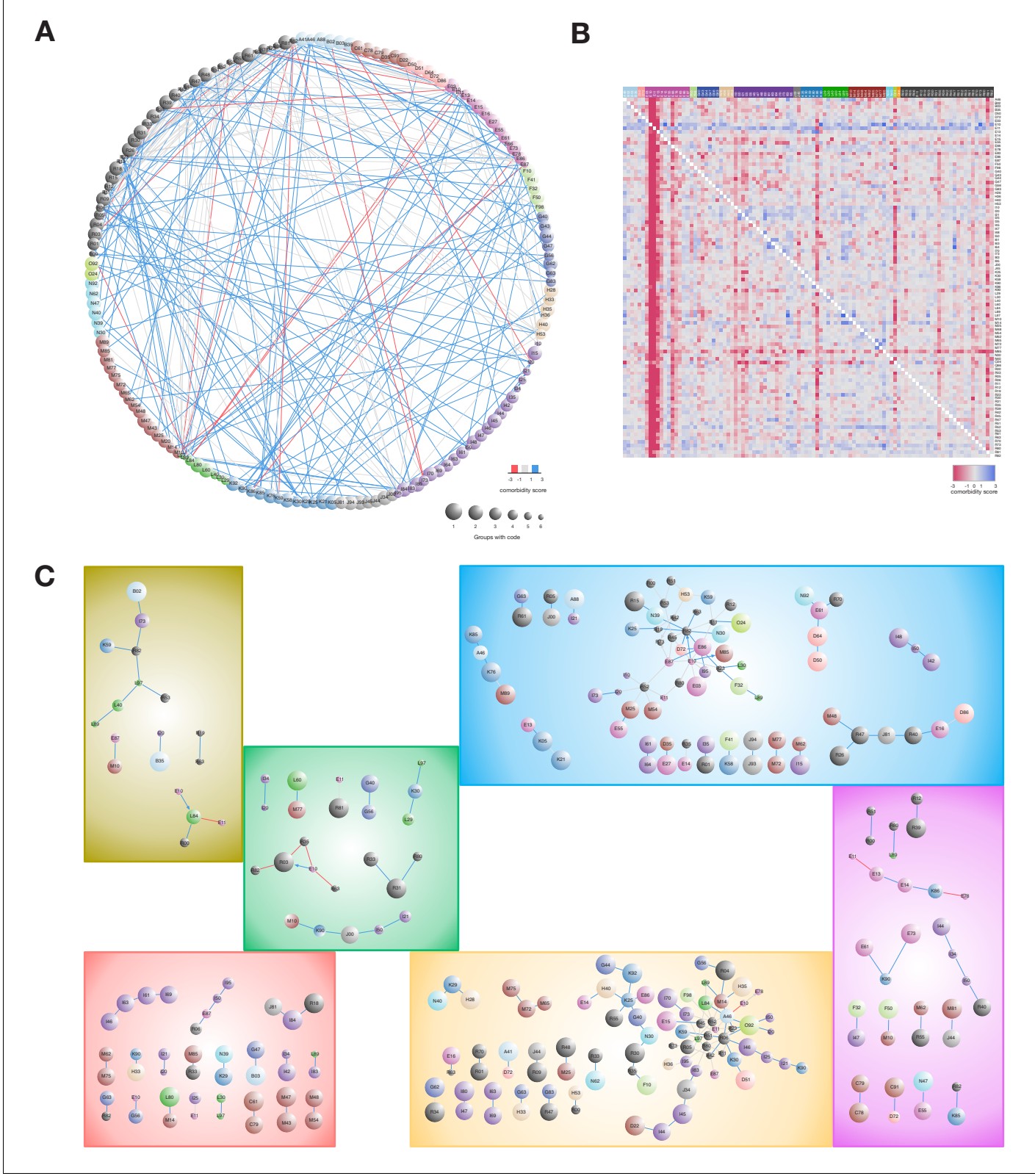

**Figure 4.** Comorbidity patterns within the six symptom groups. (**A**) Comorbidity correlations between the combined symptom groups. (**B**) Asymmetric comorbidity matrix for observing row diagnosis codes before column diagnoses. First, we calculated Bonferroni corrected p-values for diagnosis pair directionality, second, we extracted the top 100 unique diagnosis codes pairs with lowest adjusted p-values and lastly, we calculated a comorbidity score (CS) by using the log2 of observing the pair more or less than expected. The heat-map colors reflect the CS quantification. (**C**) Comorbidity pairs

*Figure 4 continued on next page*

*Figure 4 continued*

unique for each of the symptom groups. All interactions are observed significantly more (blue) or less (red) than expected (adj. p-value<=0.01). Arrows indicate that the diagnoses are observed in the particular order (Fischer's exact test with Bonferroni correction p-value<=0.01). Node size indicates in how many symptom groups the diagnosis code is observed in, ranging from one group (the diagnosis is unique for the group, largest nodes) to six groups (all groups have the code, smallest nodes).

The online version of this article includes the following source data for figure 4:

**Source data 1.** Comorbidity pattern data.

spondylopathies and soft tissue disorders. Further, these diagnoses are observed exclusively in this group and show unique disease co-occurrence patterns, for example M48-M54 (*other spondylopathies* and *dorsalgia*, CS = 1.01, adj. p-value=1.8e-04) and M43-M47 (*deforming dorsopathies* and *spondylosis*, CS = 1.54, adj. p-value=1.91e-06). One of the top ranked genetic associations for this cluster (rs76548985, p-value=1.43e-06) is *LINC00351,* associated with sporadic amyotrophic lateral sclerosis (*Xie et al., 2014*). It is worth noting that clusters 8, 22, 33, 35, 45 within symptom group five are all enriched for drugs from ATC chapter A10B: *blood glucose lowering drugs, excluding insulin* (adj. p-value<0.05), and all but cluster eight are associated with glycemic dysregulation.

Within symptom group 7, we observed two diagnosis pairs less than expected: E11-E13 (CS = −1.46, adj. p-value=1.57e-04), and K86-E78 (CS = −1.24, adj. p-value=5.46e-04). Hence, this group contains patients where T2D and *other diabetes* as well as *diseases of the pancreas* and *disorders of lipoprotein metabolism* are not given together. In contrast, I34: *nonrheumatic mitral valve disorder* is observed more often than expected together with *heart failure* (I50, CS = 1.83, adj. p-value=0.009) and *atrioventricular and left bundle-branch block* (I44, CS = 1.53, adj. p-value=0.0018). Interestingly, one of the top genetic signals for symptom group seven maps to *MIR8052* (rs6590490, p-value=3.14e-07) that has been associated with pulse pressure (*Warren et al., 2017*). Comparably, among the top genetic signals for symptom group 4, a group where are large proportion of the patients are characterized by hypotension (I95) and vertigo (R42), are *ANLN* that has been associated with systolic blood pressure (*Parmar et al., 2016*).

## Glycemic dysregulation

We evaluated five different parameters associated with glycemic dysregulation (glycemic dysregulation, hyperglycemia, check-point detection of fluctuating HbA1c levels, HbA1c level at diabetes onset and amount of HbA1c observations above diagnosis threshold for T1D and T2D [53 mmol/ mol]) and found that 2942 patients did not meet any threshold criterion, 2484 met one, 4647 two, 4057 three, 531 four, and 22 met all five criteria. The distribution of HbA1c measurements for T1D and T2D is shown in *Figure 1—figure supplement 5*. First, we investigated whether there was any difference in mean values of the 20 different biochemical tests (see Material and methods) and subsequently we applied a Kolmogorov-Smirnov test to assess how these distributions differed. We found that the means of 14 of the different biochemical tests were differently distributed between the six groups (adj. p-value<0.01) of patients with different number of dysregulation parameters, and furthermore observed a distinct difference between the not-or-slightly dysregulated patients (groups 0 to 2) and the middle-or-highly dysregulated patients (groups 3 to 5) (*Figure 1—figure supplement 6*). The group with five parameters showed no significant difference, due to the low number of patients (N = 22). The group with 3–5 parameters showed higher levels of triglyceride and HbA1c, and lower levels of sodium, urine creatinine, C-peptide, hemoglobin, diastolic blood pressure and height. Elevated levels of HbA1c, triglyceride, LDL-cholesterol and cholesterol and lower levels of HDL-cholesterol are known biochemical values associated with glycemic dysregulation and thus verified our findings.

The detection of higher levels of potassium and plasma creatinine as well as the lowered sodium, urine creatinine, hemoglobin levels indicates that these biochemical tests might be used in future prediction of glycemic dysregulation. Glycemic dysregulation is expected to cause renal problems, (identified by elevated plasma creatinine and elevated urine albumin) and hypertension, which is treated with RAS blocking agents (ACE inhibitors and angiotensin two receptor blockers) and diuretic agents, which elevate potassium and lower sodium. The treatment profile of this patient group revealed an enrichment of patients treated by RAS blocking agents in most of the clusters.

Based on these observations we considered having at least three of the parameters as the best approximation for a definition of potential glycemic dysregulation.

Using the results from the biochemical analysis, we divided the cohort in two: those with at least three parameters associated with glycemic dysregulation, and those with two or less. In the 71 clusters defined above, 21 had more than 50% patients with at least three parameters (*Figure 3*, red circles). We found 10 of the 21 clusters in symptom group 3 of which, cluster 5, 24, and 47, were enriched for poor compliance when using the SDC-custom dictionary (adj. p-value=5.9e-03, 1.9e-03 and 2.6e-02, respectively). By further investigating the enrichment of SDC-custom terms (adj. p-values≤0.05) we found that the majority of the 21 clusters had terms related to cardiovascular complications (e.g. beta blocks, ischemia, diuretics and bypass), kidney complications (e.g. nephropathy, edema and albuminuria), metabolic complications (hypoglycemia and insulin chock) and neurologic related disorders (e.g. neuropathy and loss of memory). Furthermore, all the patients in cluster 47 have schizophrenia (N = 76, adj. p-value=8e-141), and behavioral features might therefore account for the glycemic dysregulation. The same could be the case for cluster 24, in which all have epilepsy (N = 108, adj. p-value=7.6e-186).

## Genetic characterization of dysregulated patients

To assess if glycemic dysregulation is a diabetic complication or evidence of disease etiology, we further tested whether any SNPs were associated with glycemic dysregulation (n = 2,120). The five top associating signals map to *NCKAP5, CLNK, PSD3, KPNA5,* and *LINC00333* (*Supplementary file 5*), although not reaching genome-wide significance. Interestingly, two of the genes associated with schizophrenia (*LINC00333* [*Goes et al., 2015*] and *NCKAP5* [*Draaken et al., 2015*]) and *PSD3* have also been associated to traits related to urinary and blood metabolite levels, metabolic traits, and triglyceride levels (*Raffler et al., 2015*; *Teslovich et al., 2010*; *Shin et al., 2014*; *Rueedi et al., 2014*). However, none of the five top ranked genes have been previously linked to glycemic levels or diabetic dysregulation.

## Discussion

Previous studies using EHRs in diabetes research have focused on improving clinical decision making (*O'Connor et al., 2011*), clinical prediction (*Miotto et al., 2016*), patient management (*Cebul et al., 2011*), mortality risk (*Pantalone et al., 2009*; *Pantalone et al., 2010*), genetic risk factors (*Kho et al., 2012*), and subgroup identification (*Li et al., 2015*). Only the study by *Miotto et al. (2016)* used the different layers of the EHRs, aimed at predictive measures of clinical outcome. A study from the eMERGE consortium extracted phenotypes from EHR narratives by using NLP-based methods (*Kho et al., 2011*). They used EHR for phenotypic characterization of five main diseases, but a fine-grained analysis of phenotypic characterization within the diseases was not performed. Further, NLP was included only in the phenotypic determination of three of the diseases, not for diabetes determination.

Stratification and subdivision of diabetic cohorts have typically been performed on homogeneous data sets within specific diabetes types such as T1D, T2D, or gestational diabetes (*Perry et al., 2012*; *Ren et al., 2016*; *Lin et al., 2012*; *Achenbach et al., 2004*). One of the more recent stratification studies of diabetes patients is *Li et al. (2015)* that identified subtypes of T2Ds of mixed ethnicity using the structured part of EHRs. They detected three distinct subgroups that could be linked to significant SNPs through gene-disease associations in a patient-unspecific manner. Further, elevated HbA1c levels were used to explain one subgroup with microvascular diabetic complications. In contrast to the study by Li et al., we have taken the stratification and characterization several steps further both by investigating a heterogeneous diabetic cohort almost five times as large and obtaining the full comorbidity pattern and symptoms relatedness through mining of the text-narratives using both an 'exposure-oriented' and a diagnosis-based dictionary. In addition, we used the biochemical data to produce a severity classification (the five parameters of glycemic dysregulation) and integrating this with both the text-mined and assigned diagnoses, we were able to determine many different, more homogeneous groups of patients with shared symptoms and comorbidities, as well as different levels for glycemic dysregulation.

Another recent diabetes stratification study by *Ahlqvist et al. (2018)* used a data-driven approach and k-means clustering to subgroup adult-onset diabetes and characterize five subgroups

showing differing disease progression and risk of diabetes complications. However, this approach concerned only individuals with type 2 diabetes and a characterization based on six parameters (glutamate decarboxylase antibodies, age at diagnosis, BMI, HbA1c, and homoeostatic model assessment 2 estimates of β-cell function and insulin resistance), and thus clinical narratives, medication, and genetics were not used as we have done in this study.

The text mining approach used in relation to ICD-10 codes was based on level three rather than the more detailed level four since it would increase tremendously the dimensionality of the feature space. While this obviously reflects a less deep phenotyping, for a data set of this size many level four codes would be unique, likely leading to a less stable subsequent clustering and analysis. In fact, our attempt to use the much more fine-grained SNOMED-CT terminology confirmed that a data set needs to be very large for such a fine-grained vocabulary to be useful.

In this work, we deliberately excluded the primary diabetes types without complications, T1D and T2D, and thereby constructed a stratification of the cohort driven solely by comorbidities, complications, other diseases, and symptoms. However, combining different diabetic subtypes can be problematic, since their etiologies differ and disease progression is different across diabetes types, treatment, compliance and lifestyle (*Adeghate et al., 2006*). Our focus was not to characterize specific comorbidity-related groups within a certain diabetes type, since extensive epidemiological studies of this kind have been done previously. Instead, we focused on the diabetes continuum with the aim of investigating whether it was possible in an unsupervised manner to detect relevant and meaningful diabetic subgroups by comorbidities, symptoms, or level of glycemic dysregulation. Further, we detected novel biochemical and genetic candidates that might relate these to the different cohort subdivisions, such as shared symptom patterns for phenotypically similar patients and the level of glycemic dysregulation. These biochemical and genetic candidates could be potential risk factors for additional complications, especially concerning glycemic dysregulation, that could be verified by further experimental studies. As the cohort is enriched for sicker patients with diabetes melitus complications the features and the overall grouping described would not necessarily be the same if another cohort dominated by prediabetes individuals would have been analyzed.

Despite our focus on the phenotypic variation among diabetes patients, the stratification is restricted by the limited coverage of the genetic data, which lowers the power considerably. We were able to obtain genetic data for 2337 patients, of whom 2125 remained after quality control and stratification. Hence, only 14% of the patients in our final cohort had descriptive genetic information.

By adding biochemical, prescription, and genetic data we observed that the clusters were significantly different from each other on parameters other than comorbidities. By including the text narratives of the EHRs we were able to capture diagnoses that in another context would be considered as a primary diagnosis, for example epilepsy, schizophrenia and cerebral palsy. These diagnoses are not known comorbidities of diabetes but can influence the treatment and management of the diabetes patient. For instance, we observed that all patients in cluster 47 had schizophrenia, which could influence their compliance since the cluster was associated with glycemic dysregulation. We determined this when assessing the level of glycemic dysregulation and found that this cluster indeed showed a high number of patients with at least three parameters for glycemic dysregulation. However, a more in-depth analysis is required to clarify whether the glycemic dysregulation is due to the behavioral effects of schizophrenia, underlying genetic variants, adverse drug reactions due to polypharmacy, or other variables.

Despite our data from both assigned and text-mined diagnoses, misdiagnoses can occur, and we performed a manual inspection of randomly selected EHRs to establish the validity of the data. Furthermore, we observed some patients assigned with different diabetes types, for example first assigned with T1D and later with T2D, and vice versa. Inspecting the biochemical values of GAD65 autoantibodies and comparing them to the primary diagnosis type we found 182 T2D assigned individuals to have GAD65 levels above 10 IU/ml, possibly indicative of T1D or LADA; however, these individuals were not significantly enriched in any cluster. We also observed 621 individuals with GAD65 levels below 10 IU/ml, which is consistent with known late-term effects of T1D (results not shown). An in-depth temporal analysis of these patients with mixed diabetes types could be interesting and integrating biochemical as well as genetic variation data could elucidate which, if any, phenotype might be the most accurate.

In this study, we have used data from a unique cohort of 14,017 patients with diabetes, of which 12,866 had been diagnosed with either T1D or T2D. Integrating the assigned and text-mined ICD-10 and SDC-custom diagnoses, an MCL clustering was carried out which resulted in 172 unique clusters. Of these, 71 had at least 50 patients, which were subsequently divided into groups with shared symptoms. Investigating the complication enrichment and comorbidity patterns in the clusters and symptom groups we detected clusters described by specific disorders such as hypothyroidism, schizophrenia, and functional intestine disorder as well as unique comorbidity interaction patterns both with and without temporal significance. An interesting approach could be to extend the temporal analysis to investigate how disease progression within and between clusters and symptoms groups develops for multiple diagnoses. This could be done with a trajectory-based approach as done recently by *Jensen et al. (2014)*.

## Materials and methods

### EHR data

All data originate from the Steno Diabetes Center Copenhagen (SDCC), a specialized diabetes hospital in the Capital Region of Denmark. In Denmark patients with type 1 diabetes (T1D) are followed in hospital outpatient clinics such as SDCC, and the T1D patients studied comprise 35% of all adult patients with T1D in the Capital Region of Denmark. Patients with type 2 diabetes (T2D) are referred from primary care for treatment optimization, typically for a period of six to twelve months. When treatment goals are reached, and they have no diabetic complications, they are referred back to general practice. Patients needing intensive control and treatment, because of micro- or macrovascular complications, are offered life-long follow-up at the SDCC. At any time, approximately 2000 patients with complicated T2D are followed at the SDCC. Generally, the patients registered in the SDCC electronic patient records are representative of Danish patients with T1D and the 10% most complicated patients with T2D (*Jørgensen et al., 2016*). Moreover, the patient followed at SDCC are comparable to patients followed in all Danish hospital diabetes outpatient clinics in terms of distribution of age and duration of diabetes. The data comprise all communications and contacts recorded at the hospital over a period of 19 years (1993–2012) for 14,017 patients. This includes, primary diagnoses, prescriptions and laboratory tests, 1.2M clinical narrative entries, 420 different types of laboratory tests with 4.15M laboratory measurements and a total number of 440,555 drug prescriptions. On average, each patient had 85 clinical narratives with an average length of 34 words (212 characters). In addition, genetic data from several research projects have been linked to the patients and added to the EHRs.

### Text-mining dictionaries, tagging and corpus matches

An in-house developed framework for mining Danish text was used for the analysis (*Roque et al., 2011*; *Eriksson et al., 2013*). The algorithm tags words in the text narratives in a named entity recognition (NER) fashion based on supplied dictionaries. In this study, we used two main dictionaries: The International Classification of Disease version 10 (ICD-10) truncated to level 3 (e.g. E10: *Type 1 Diabetes*), and a complementary 'exposome-oriented' dictionary (SDC-custom). The latter holds terms related to diabetes specific subtypes (e.g. MODY and LADA), complications (e.g. the different severities of neuropathy, retinopathy and nephropathy), treatments and examinations (e.g. gastric bypass, renography, and beta blockers), lifestyle and lifestyle related disorders (e.g. obesity, exercise level, smoking), and compliance. The SDC-custom dictionary was developed in collaboration with physicians at the SDCC (see *Supplementary file 6* for a translated and condensed version). The Danish ICD-10 version currently contains roughly 20,000 unique descriptions of clinical concepts, each with a unique ICD-10 code.

The NER used for dictionary matching, in addition, performs lemmatization and de-latinization of tagged words, accounts for language negations or subject negations (e.g. 'the patient's mother had retinopathy'), and performs fuzzy matching with a Hamming distance of 1 (e.g. 'diabtes' is transformed to its correct spelling 'diabetes'). A thorough explanation of the algorithm is provided (Simon et al., 2019, manuscript in preparation). Other details, for example on 'negation scope', that is the position of negations relative to the negated term in Danish, have been published previously (*Thomas et al., 2014*).

Running the text-mining algorithm (Simon et al., 2019, manuscript in preparation) on the SDCC corpus with the two dictionaries (ICD-10 and SDC-custom) recognized 1,028,593 entities from the dictionaries in 12,504 patients (80.5% of the entire corpus). None of the remaining patients had any non-trivial match between the dictionaries and EHR narratives. The two dictionaries shared some general terms, for example T1D and T2D; these duplicate matches were removed and 941,087 unique code matches remained. Of these, 267,404 were fuzzy matches representing 4181 unique variants. The variants were manually validated, resulting in removal of 10,952 (4.1%) matches. After removal of negated sentences (n = 255,302) 594,600 code-to-text matches in 12,467 patients were left.

## Patient phenotype vectors from assigned and text mined codes

The structured ICD-10 codes assigned to patients during their contact with SDCC were extracted from the EHRs, along with all ICD-10 codes captured by mining the text parts of the EHRs. The two ICD-10 lists were combined, but to prevent the primary, assigned diabetes types from dominating the patient stratification, diagnosis codes for diabetes without complications (E10 and E109, in total 3740 codes, and E11 and E119, in total 3624 codes) were removed. Approximately 8% of the assigned codes were removed in this way. The list of codes and their frequencies for each patient were transformed using the BM25 weighting scheme (*Robertson and Walker, 1994*), which scores a code *c* in patient *P*, accounting for the code frequency in all patients, frequency of the codes in the patient (document frequency), and number codes in the patient record (document length), see *Equation 1*.

$$Score\,(p,c) = \sum_{i-1}^{n} IDF(c_i) \cdot \frac{f(c_i,p) \cdot (k_1 + 1)}{f(c_i,p) + k_1 \cdot \left(a - b + b \cdot \frac{|p|}{|p_{ave}|}\right)} \tag{1}$$

Here, $IDF(c)$ is the inverse document frequency for the code $c$ computed as

$$IDF(c_i) = \log \frac{N - n(c_i) + 0.5}{n(c_i) + 0.5}$$

With $N$ being the total number of patients and $n(c)$ the number of patients with a given code $c_i$, and the term $f(c_i, p)$ is the frequency of code $c_i$ in patient $p$. The number of codes associated with each patient vector, $P$, is given by the length of the vector, $|p|$, and the average number of codes in the entire corpus is $|p_{ave}|$. Finally, $b$ and $k_1$ are free parameters that determine to what extent document length is considered ($b$) and how much the scoring equation resembles a normal TF-IDF ($k_1$), respectively. The value of $b$ was set to 0.75 and does not fully account for the document length ($b = 1$) and $k_1$ was set to 1.2 giving a low resemblance of TF-IDF ($k_1 \to \infty$).

## Clustering patients from Cosine similarities

All patients were clustered using their pairwise cosine similarities calculated from the BM25 transformed code vectors. A cosine similarity $\geq 0.5$ was set as a cut-off prior to clustering, to minimize the number of edges in the subsequent patient network. To increase the variance of the cosine similarities, these were scaled from the interval 0.5–1 to 10–100. We wanted to do a network based clustering and therefore used Markov Clustering (MCL) with the inflation parameter set to 1.2 and rest left as default (*Van Dongen, 2000*). Different inflation parameters were tested and evaluated based on the efficiency, mass fraction, and area fraction parameters.

## Grouping clusters in symptom related groups

We organized the clusters into symptom groups based on the frequency of their symptom codes using ICD-10 chapter XVIII level 1, for example R50-69: *General symptoms and signs*. We used a Euclidean distance and applied a hierarchical clustering using Ward.D as the agglomeration method since we wanted to expose the hierarchical relationship amongst the clusters. The entire analysis was performed using R (version 3.2.1).

## Enrichment analysis of diagnosis codes

The MCL clusters were tested for ICD-10 and SDC-custom codes found more often than expected, using a binominal test while correcting for sex and birth decade. The metadata such as average age, days at SDCC, and diabetes duration (from the date of diabetes diagnosis until the end of the study) were calculated, and further p-values for each cluster were obtained using a Wilcoxon test against the remaining clusters. In both analyses, p-values were adjusted using Benjamini-Hochberg correction for multiple testing, where a p-value$\leq$0.05 was considered significant.

## Comorbidity patterns for diagnosis pairs

We performed three independent analyzes without considering the clusters by applying Fischer's exact tests to obtain p-values for all diagnosis pairs within the SDCC corpus: 1) p-values for observing the codes together, 2) p-values for observing diagnosis A prior to diagnosis B, and 3) p-values for observing diagnosis B prior to diagnosis A. P-values from the three different sets were adjusted using Bonferroni correction for multiple testing, and the pairs were subsequently ranked based on these values. To detect whether the pairs were observed more together than expected we applied a comorbidity score as described in *Roque et al. (2011)*. For the temporal pairs, we also applied an ANOVA test to investigate whether any of these pairs were unique for a symptom group. All p-values were corrected for multiple testing, and an adjusted p-value$\leq$0.05 was considered significant.

## Robustness of the MCL generated clusters

To assess quantitatively the stability of the clusters generated, we constructed various diluted and shuffled realizations of the similarity network used as input to the MCL algorithm. We used a reference clustering similar to the clustering presented in *Figure 2B* (either by including the patients in the 71 clusters or all patients). The diluted versions were generated by randomly deleting edges with a probability of $\alpha$, whereas the shuffled realizations were created by shuffling edges between nodes (patients) as described earlier (*Karrer et al., 2008*). The latter produces a network where the number of edges and vertices are unchanged. An $\alpha$ of zero leaves the reference network unchanged, while a value of 1 leads to a complete randomization of the similarity network. Each of these randomizations of the input were repeated five times for various values of $\alpha$ in the range 0–50% and used as input for the MCL algorithm. The resulting clustering's were then compared to the reference clustering by means of the Variation of Information measure (VI) (*Meilă, 2007*) and plotted as function of increasing values of $\alpha$ (see *Figure 2—figure supplement 5*). The figure includes two horizontal lines corresponding to the value that the VI would take if we were to randomly assign 10% and 20% of the vertices to different random clusters, respectively. This analysis showed that the clustering is stable in relation to removing edges, which is evidence that the cosine metric-based cutoff used does not change the overall structure of the clustering. The shuffling is a more impactful randomization, however despite this, we can still shuffle around 10% of the edges and still retrieve 90% of the patients in the groups of the 71 reference clusters.

## Quantitative assessment of glycemic dysregulation

Glycemic dysregulation was assessed for each patient by evaluating five different parameters. The first two parameters were obtained using the SDC-custom code for dysregulation (sdcL03) and the ICD-10 codes for hyperglycemia (R73 and E89). The remaining three were found by analyzing longitudinal measurements for glycated hemoglobin (HbA1c). Due to a large variation in both the number of measurements and their frequency, HbA1c values were pre-processed. We divided the HbA1c measurements for each patient into segments containing a minimum of five values, spanning a time interval of at least three months (equivalent to the functional lifetime of red blood cells). In total 10,112 patients had HbA1c measurements that fulfilled the criteria, and the subsequent analyses were performed on this sub-population.

We performed three analyses on the longitudinal pre-processed HbA1c data for each patient: 1) a Bayesian analysis of change point detection to find potential peaks of HbA1c values in a patient, 2) analysis of mixed effects models to estimate the HbA1c value at diabetes onset, and, 3) analysis of the frequency of values in different HbA1c bins (e.g. general level for diagnosing T1D or T2D, the critical interval for hyperglycemia etc.) to appoint an HbA1c severity score.

## Laboratory test data

The laboratory tests were longitudinal data such as blood pressure measurements and biochemical analyses of blood and urine samples, and each test was assigned a unique identifier using the NPU-terminology, which is the recommended administration and communication measure of laboratory tests in Denmark (*Petersen et al., 2012*). In our data, several laboratory tests had an SDC identifier, being from local laboratory facilities at SDCC. Both test IDs, NPU and SDC, were analyzed separately, despite sometimes measuring the same biochemical variables.

In total, 420 different physiological tests were performed across 14,847 patients from the entire corpus. Measurements within and between tests were unbalanced with no general system in measurement interval, frequency, or number of patients who had a test taken. Due to this lack of systematic coverage, only tests that were performed on at least 75% of the entire corpus (10,788 patients) were analyzed (26 tests). However, the test for C-peptide (NPU18004) was also included as it was available for 74.9% of the cohort and is widely used to distinguish T1D and T2D. Measurements outside the biological reference interval for a given test, that is HbA1c measurements below 15 mmol/mol and above 184 mmol/mol, were removed, and for each patient the mean, median and standard deviation for each test with continuous values (20 of the 26 tests) were calculated. If the data was not normally distributed for a test we log-transformed it and normalized all values to mean = 0 and SD = 1. All analyses after assigning patients to clusters were performed on the 10,788 patients.

We applied a MANOVA to test if means among the three different patient groups (clusters, symptom groups or patients being dysregulated) were significantly different, and a Kolmogorov–Smirnov test was applied to investigate whether the distribution of the sample means in the patient groups were significantly higher or lower than means in the remaining groups. All p-values were adjusted using Bonferroni correction for multiple testing, and an adjusted p-value$\leq$0.05 was considered significant.

## Drug prescription data

Prescription data was available for 12,147 patients with a total number of 440,555 drug prescriptions. Drug compounds were identified by the ATC classification system, which is divided into groups at five different levels. In this study, we summarized the data using ATC-codes at level three and four: *chemical and pharmacological* and *therapeutical*, respectively.

From the initial set of prescriptions, we manually reviewed 104 drugs which did not have an ATC code in the EHR or were mapped to more than one ATC code. In addition to the manual review, pro.medicin (www.pro.medicin.dk, accessed October 2018) was used to map drug names to their corresponding ATC code. The SDCC prescription data and the WHO Collaborating Centre for Drug Statistics Methodology (www.whocc.no, accessed October 2018) were used for crosschecking. We performed Fisher's exact test to investigate prescription enrichment (3rd level of the ATC classification) in clusters with at least 50 patients. The p-values were adjusted using Benjamini–Hochberg correction for multiple testing, and an adjusted p-value$\leq$0.05 was considered significant.

## Genomic characterization

A total of 2290 patients with T2D and 1028 patients with T1D from SDCC were genotyped separately using the HumanOmniExpress (24v1) array from Illumina as previously described (*Charmet et al., 2018*; *Steinthorsdottir et al., 2014*). Genotypes were called using GenomeStudio, and imputed separately using the Haplotype Reference Consortium (HRC) imputation panel (*McCarthy et al., 2016*). Prior to imputation, the two datasets were filtered to retain only high-quality samples/SNPs (sample call rate $\geq$98%, no mislabeled sex, no ethnic outliers, heterozygosity within 2 SD from the mean, SNP call rate $\geq$98%, no monomorphic SNPs, no Hardy–Weinberg disequilibrium outliers). After imputation, SNPs with minor allele frequency (MAF) <0.01, more than 20% missingness, R square less than 0.30, and duplicate SNPs were removed, and the two datasets were merged retaining only variants common to the two sets. After merging, relatedness between all individuals were calculated and close relatives were excluded. Of the 3318 patients, 2337 had EHR information and could be mapped to clusters. In total 2125 patients passed quality control and were taken forward for genomic characterization. Logistic regression was used to test for genetic differences (PLINK 1.90 beta, https://www.cog-genomics.org/1.9) between the different groups of interest (clusters and symptom groups) and linear regression was used to evaluate the SNPs impact on

dysregulation. Cases were defined as all individuals in a given cluster/symptom group, and controls as all individuals not belonging to the respective cluster/symptom group. Glycemic dysregulation was defined as a score ranking from 0 (low) to 5 (high) based on five dysregulation parameters (see section on glycemic dysregulation). All analyses were adjusted for age and sex. The test statistics were adjusted for inflation (population stratification) using the three first principal components estimated using the –pca function in PLINK. Genetic associations were defined based on data derived from the EBI GWAS catalog version 1.0.1 (http://www.ebi.ac.uk/gwas/) unless otherwise stated. A p-value less than 5*10–8 was considered genome-wide significant.

## Additional information

### Funding

| Funder | Grant reference number | Author |
| --- | --- | --- |
| Danish Council for Strategic Research | 0603-00321B | Søren Brunak |
| Innovation Fund Denmark | 5153-00002B | Søren Brunak |
| Novo Nordisk Foundation | NNF14CC0001 | Søren Brunak |
| Novo Nordisk Foundation | NNF17OC0027594 | Søren Brunak |

The funders had no role in study design, data collection and interpretation, or the decision to submit the work for publication.

### Author contributions

Isa Kristina Kirk, Christian Simon, Conceptualization, Resources, Data curation, Software, Formal analysis, Validation, Investigation, Visualization, Methodology, Writing—original draft, Project administration, Writing—review and editing; Karina Banasik, Formal analysis, Methodology, Writing—original draft, Project administration, Writing—review and editing; Peter Christoffer Holm, Amalie Dahl Haue, Formal analysis, Methodology, Writing—review and editing; Peter Bjødstrup Jensen, Formal analysis, Supervision, Methodology, Writing—review and editing; Lars Juhl Jensen, Conceptualization, Data curation, Formal analysis, Supervision, Methodology, Writing—original draft, Writing—review and editing; Cristina Leal Rodríguez, Methodology, Writing—review and editing; Mette Krogh Pedersen, Robert Eriksson, Data curation, Methodology; Henrik Ullits Andersen, Conceptualization, Resources, Data curation, Validation, Writing—review and editing; Thomas Almdal, Oluf Pedersen, Conceptualization, Resources, Data curation, Writing—review and editing; Jette Bork-Jensen, Niels Grarup, Data curation, Writing—review and editing; Knut Borch-Johnsen, Conceptualization, Resources, Supervision, Project administration, Writing—review and editing; Flemming Pociot, Resources, Data curation, Methodology, Writing—review and editing; Torben Hansen, Conceptualization, Resources, Data curation, Supervision, Writing—review and editing; Regine Bergholdt, Conceptualization, Resources, Data curation, Methodology, Project administration, Writing—review and editing; Peter Rossing, Conceptualization, Resources, Data curation, Supervision, Methodology, Writing—original draft, Project administration, Writing—review and editing; Søren Brunak, Conceptualization, Resources, Data curation, Funding acquisition, Methodology, Writing—original draft, Project administration, Writing—review and editing

### Author ORCIDs

Søren Brunak https://orcid.org/0000-0003-0316-5866

### Decision letter and Author response

Decision letter https://doi.org/10.7554/eLife.44941.sa1
Author response https://doi.org/10.7554/eLife.44941.sa2

## Additional files

### Supplementary files

• Supplementary file 1. Statistics for the metadata.

• Supplementary file 2. Statistics for the physiological tests.

• Supplementary file 3. Enrichment of drug prescriptions.

• Supplementary file 4. Enrichment of ICD-10 and SDC-custom codes.

• Supplementary file 5. The five top ranked independent genetic associations for individual clusters, symptom clusters, and dysregulation.

• Supplementary file 6. The SDC-custom dictionary.

• Transparent reporting form

### Data availability

All data generated or analysed during this study are included in the manuscript and supporting files except for the raw person sensitive electronic health record data due to confidentiality requirements.

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
