## [Decision Letter]

**Acceptance summary:**

The paper describes the stratification of a diabetes cohort based on characteristics extracted from the medical records of a rather homogenous population. These characteristics include other diagnoses and lifestyle factors. The clusters of patients, obtained with an unsupervised Markov clustering method, represent characteristic longitudinal glycemic dysregulation patterns that include the temporal order of comorbidities, as well as genetic features, i.e. SNPs close to some of the known diabetes related genes.

The work opens the doors to the study of the molecular basis of the classical etiological subtypes of diabetes at the light of their relations with the clusters based on comorbidity relationships and symptoms.

**Decision letter after peer review:**

Thank you for submitting your article "Linking glycemic dysregulation in diabetes to symptoms, comorbidities and genetics through EHR data mining" for consideration by *eLife*. Your article has been reviewed by three peer reviewers, and the evaluation has been overseen by a Reviewing Editor and Naama Barkai as the Senior Editor. The reviewers have opted to remain anonymous.

The reviewers have discussed the reviews with one another and the Reviewing Editor has drafted this decision to help you prepare a revised submission.

Summary:

The biomedical goal of the paper is to identify subgroups of diabetes patients. The study is based on a large set of medical records from patients with T1D or T2D. The methodology produces 6 large clusters (MCL) combining structure information (ICD-10 level 3 codes) and symptoms extracted from the free text. The analysis shows enrichment in clinically relevant groups and relations with other diseases. The paper includes the specific analysis of glycemic deregulation, its relation with comorbidities and with previously identified genes/SNPs.

Essential revisions:

The topic is relevant and timely. The technical contribution is at the leading front of the analysis of medical records analysing structured and free text information.

On the critical side there are limitations in clustering analysis, as well as in the use of limited ICD third level data. These serious criticisms have to be addressed completely in a revised version. There are also a number of other issues related with the origin of the data (temporal coverage of the cases) that need to be addressed in detail in the text.

Finally, it is not completely clear what are new results and what is validated (and how). Even, if these are common problems in many large scale studies, it is important to address them clearly in the paper. This point is related with the criticisms on the unclear goal of the paper: technical or medical? The comments of the reviewers reflect that given the situation, you will have to think if you want to prioritise the novelty of the approach instead of making medical claims that might be difficult to sustain.

We ask you to address the following;

More details on the key points above:

1) Circularity

The analysis strategy and reasoning appears to be circular and there are not provided measures of independence between codes, factors, elements, etc. that are introduced or measured at each step. In particular, there are used two vocabularies (which obviously expand the number of "codes" assigned to patients, as it should be; first section Results is not novel) that gave approximately 940,000 codes as explained in Materials and methods. However, it is expected that many of these codes are in fact correlated, non-independent, but instead they are all used to generate patient clusters; for example, in several instances it is emphasized that codes from primary diabetes diagnoses are not used in clustering/analyses, but it is well known a priori that many other codes are significantly correlated with the diagnoses (it is therefore not unexpected that patient clusters differentiate diabetes subtypes). Therefore, it would also not be unexpected that clusters differentiate glycemic levels. In fact, the 71 clusters are defined using distances from ICD-10 chapter XVIII symptoms (subsection “Enriched comorbidity and symptom patterns in diabetes patient clusters”, first paragraph), which includes examination of blood parameters. Again, latter the clusters are defined based on comorbidities, which presumably many are linked to the original codes. Another example of this point is the identification of a "schizophrenia" cluster (subsection “Glycemic dysregulation”, last paragraph), which is expected if the corresponding codes are included. I agree that an observation of differential glycemic levels in this clusters may be interesting, but then first it should be evaluated the overall correlation among the thousands of codes used in the study, and also then the study is limited to assess co-occurrences of terms/codes.

One possible way out could be the comparison results of structured vs. structured+unstructured features.

2) Clustering

As overall conclusion, Discussion section includes the statement that the results show "many different and homogenous groups of patients", but it has to be formally demonstrated that the groups are independent, that they are many more than randomly expected by permuting the same cohort, and that they are more homogenous that some value.

Additionally, questioning the clusters stability, would be necessary to provide a sensitivity analysis on the clusters to illustrate the stability.

3) ICD codes

Truncating the codes to three digits is akin to adding noise by losing information. Please justify your rationale behind truncating the codes. It'll be good if this issue could be addressed in the Discussion or one of the limitations of the work.

4) Data limitations and temporal series

First, there might be problems with the time-dependent comorbidities in a given cluster. There are multiple reasons: (1) the length of the observational window before and after considered in a given cluster is not talked about/defined. This can impact the enrichment of symptoms and hence all the downstream analysis; (2) length observational windows is bound to be different in different clusters, which may introduce the bias in computing comorbidities; (3) it is not clear from the analysis if there exist any relation between the observational window and cluster size.

Indeed, it is not clear how temporality is defined. The heatmap presented in Figure 2B does not help either. Specifically, what is the index date, relative to which pre and postcodes were identified? It is quite likely that the index date would vary for patients, therefore, how does that variability was accounted? Please explain this paragraph in more detail and if possible, then provide a schematic diagram representing temporality and how it was considered in this analysis. In addition, in the fourth paragraph of the subsection “Comorbidity pairs and patterns within symptom related clusters”, it appears the time window before and after is likely to bias the estimation of T2D and elevated blood glucose level. More clarity on this will be helpful.

In any case, a better explanation is needed about the length of the observational window before and after used to compute comorbidities. The length of the observational window (before and after in a given cluster) is bound to confound the disease association. Was this length kept similar for all the clusters? If not, then it is very much possible that certain cultures are (just based on cluster size) are likely to be enriched in certain symptoms. Please provide an in-depth analysis on this.

5) Additional data

Please briefly explain the type of genetic data is included and at what level of omics.

The candidate genetic and biochemical markers of glucose regulation could be of interest if replicated. However, could these have been discovered without the clustering e.g. using simpler regression methods?

It is unclear how the clusters segregate in their genetic basis, which could then support the identification of disease subtypes. The question is not as much which genetic variants are associated with a given cluster, but if the effect estimations are statistically different between clusters, which would then favor the existence of subtypes. It is also unclear if the association signals are genome-wide significant in any case.

[Editors' note: further revisions were requested prior to acceptance, as described below.]

Thank you for resubmitting your work entitled "Linking glycemic dysregulation in diabetes to symptoms, comorbidities and genetics through EHR data mining" for further consideration at *eLife*. Your revised article has been favorably evaluated by Naama Barkai as the Senior Editor, a Reviewing Editor, and two reviewers.

The manuscript has been improved but there is still some confusion about the significance of the resulting clusters, significance of the selected thresholds and the subsequent biological interpretation, as outlined below by the first reviewer. The final version should clarify the exploratory nature of the work and the limits of the statistical reliability, as pointed out by the second reviewer.

Reviewer #1:

The authors have done a perceptible effort trying to answer the major concerns initially raised, and I personally appreciate the clarifications and additional data provided.

However, I am still not convinced that the main results are robust enough to support the overall message, in particular regarding the title: "Linking glycemic dysregulation.…to symptoms, comorbidities and genetics…".

First, the issue of "circularity". I might have over-looked the original Discussion that mentioned that primary and secondary diabetes diagnoses were excluded, or my initial comment might have been unclear, but I was referring to the expectation that clusters should obviously identify major known diabetes types; in particular if dozens of diagnosis codes (some of them correlated) and extensive text vocabulary is analyzed. The point is not if the clusters identify what it is already known, but if they identify new etiology/subtypes. Of note, the following example is somewhat inaccurate: "… triple negative breast cancer patients share many features with ER-positive patients for example." Indeed TNBC and ER+ cases do not share cell-of-origin, prognosis, tissue-metastasis preference, neither therapeutic approaches.

My reasoning above is then linked to the title issue: the study of glycemic perturbation is presented in one section (“Glycemic dysregulation”) and remains unclear which clusters, symptoms, are robust regarding statistical differences for "glycemic perturbation" from the others. The thresholds of 0-2, 3-5 parameters, and then the clusters (21 with 50% patients 3 parameters…) appear to be arbitrary. The use of sentences with the following grammatical constructions may not help to understand the analyses: "distinct differentiation…dysregulation parameters". Next, glycemic perturbation is linked to genetic risk in the last Results section, but as initially raised, it is unclear what means "top association signals"; are these significant genome-wide, are they statistically different between the clusters/symptoms? Are cluster-risk interactions significant? Honestly, this remains unclear to me and I hope I am not overlooking to any data. I agree with the authors on their final response to the question of "circularity": 1) To which extent does it reproduce the classification made by doctors? 2) Does it allow us to make a more specific subgrouping of patients?

I believe that the lack of specific subgroups with statistically defined differences across the three features included in the title is evident from the Abstract, which does not detail any result: it just includes a general message of what the study has found as a last sentence.

Reviewer #2:

I am satisfied with the revision of the article. I feel that the questions I raised have been addressed. Limitation in the use of 3-digit ICD9 codes remains, but the response from authors and subsequent changes made in the manuscript are appropriate.

This study may serve as a starting point for more in-depth analysis in subsequent analysis.

---

## [Author Response]

Essential revisions:[…]1) CircularityThe analysis strategy and reasoning appears to be circular and there are not provided measures of independence between codes, factors, elements, etc. that are introduced or measured at each step. In particular, there are used two vocabularies (which obviously expand the number of "codes" assigned to patients, as it should be; first section Results is not novel) that gave approximately 940,000 codes as explained in Materials and methods. However, it is expected that many of these codes are in fact correlated, non-independent, but instead they are all used to generate patient clusters; for example, in several instances it is emphasized that codes from primary diabetes diagnoses are not used in clustering/analyses, but it is well known a priori that many other codes are significantly correlated with the diagnoses (it is therefore not unexpected that patient clusters differentiate diabetes subtypes). Therefore, it would also not be unexpected that clusters differentiate glycemic levels. In fact, the 71 clusters are defined using distances from ICD-10 chapter XVIII symptoms (subsection “Enriched comorbidity and symptom patterns in diabetes patient clusters”, first paragraph), which includes examination of blood parameters. Again, latter the clusters are defined based on comorbidities, which presumably many are linked to the original codes. Another example of this point is the identification of a "schizophrenia" cluster (subsection “Glycemic dysregulation”, last paragraph), which is expected if the corresponding codes are included. I agree that an observation of differential glycemic levels in this clusters may be interesting, but then first it should be evaluated the overall correlation among the thousands of codes used in the study, and also then the study is limited to assess co-occurrences of terms/codes.

There are several separate issues here, some relate to the interdependencies of terms in medical ontologies and some to how such terms can be used to describe subgroups of patients that share features (stratified medicine).

First, the goal in patient stratification is not to find features which are mutually independent, but to identify subgroups in larger patient populations using observable features (e.g. from physical examinations, laboratory tests, genetics or images). However, across subgroups these will typically not be unique or independent feature-wise. The entire medical profession is about combining absent or present features (also those which are mutually exclusive) in ways such that diagnoses, and diseases can be disambiguated. The medical profession is based on the use and recording of these features, and the secondary use of EHR data as in our work is also based on that premise. We do not aim to construct a new ontology with independent features; moreover, such a goal is presumably entirely unrealistic.

Second, patient subgroups are almost always interrelated, triple negative breast cancer patients share many features with ER-positive patients for example. The nested hierarchy we present with its six overall groups and additional subgroups will be interrelated in some way, especially as we go for characterizing all diabetes patients (in a continuum).

The main comment here relates to circularity, and it seems that we have done a bad job in explaining in the Results section why we have left out the primary and secondary diabetes diagnoses A/B. This was however, already included in the Discussion and perhaps the reviewer overlooked that. Contrary to cancer for example, where the ICD-10 diagnoses are quite reliable and highly detailed (lung cancer is not typically confused with liver cancer even if a patient might have both), the primary codes in a systemic, multi-organ disease like diabetes are used extremely broadly and in a fuzzy way, as the knowledge on robust diabetes subtypes and their characteristics in the context of comorbidities is quite limited. This is well-known in the diabetes EHR data mining literature (for example in the work from Vanderbilt). We have now clarified further (Results section “Comorbidity clustering based on text-mined and assigned diagnosis codes”) that we do not want the clustering to be driven by the unreliable, broad, less etiology-relevant codes (E10 and E11) from the ICD endocrinology chapter, but rather by more objectively observed symptoms, other diseases and lifestyle features. It should not come as a surprise, and it is not a sign of circularity, that there is a tendency to segregate into what is today known as type 1 and type 2 diabetes, as these overall subgroups are well established and likely real. It is the further subgrouping and the subgrouping across all diabetes patients that is the result here, not that we rediscover subtypes that have been established for a long time (since the 1920s). We have now made this much clearer in the text. The diabetes diagnoses, e.g. E10 and E11, are obviously assigned by doctors based on the same measurements and observations that we (via the text mining) use as input features. If one were to train a supervised machine learning algorithm to assign diagnoses based on these features, this would obviously be circular. However, this is not what we do. Instead, what we do is to use a non-supervised clustering approach that stratifies the patients in a purely data-driven manner. This allows us to answer two interesting questions: 1) To which extent does it reproduce the classification made by doctors? 2) Does it allow us to make a more specific subgrouping of patients?

One possible way out could be the comparison results of structured vs. structured+unstructured features.

We agree that this is highly relevant and have already in Figure 1 made such a comparison at the ICD-10 level 3. Perhaps the reviewer overlooked that.

2) ClusteringAs overall conclusion, Discussion section includes the statement that the results show "many different and homogenous groups of patients", but it has to be formally demonstrated that the groups are independent, that they are many more than randomly expected by permuting the same cohort, and that they are more homogenous that some value.Additionally, questioning the clusters stability, would be necessary to provide a sensitivity analysis on the clusters to illustrate the stability.

We fully agree that it is highly relevant to include a quantitative, robustness analysis of the clustering. To assess quantitatively the stability of the clusters generated, we constructed various diluted and shuffled realizations of the similarity network used as input to the MCL algorithm. We used a reference clustering similar to the clustering presented in Figure 2B (either by including the patients in the 71 clusters or all patients). The resulting clustering’s were then compared to the reference clustering by means of the Variation of Information measure (VI) (1, 2) and plotted as function of increasing values of α (see Figure 2—figure supplement 5). The figure includes two horizontal lines corresponding to the value that the VI would take if we were to randomly assign 10% and 20% of the vertices to different random clusters, respectively. This analysis showed that the clustering is stable in relation to removing edges, which is evidence that the cosine metric-based cutoff used does not change the overall structure of the clustering. The shuffling is a more impactful randomization, however despite this, we can still shuffle around 10% of the edges and still retrieve 90% of the patients in the groups of the 71 reference clusters. We have added these references:

1) Meilă, 2007. Comparing clusterings—an information based distance. Journal of multivariate analysis, 98(5), pp.873-895.

2) Van Dongen, S., 2000. Performance criteria for graph clustering and Markov cluster experiments. Technical Report, Centre for Mathematics and Computer Science, Amsterdam, The Netherlands.

3) Karrer, Levina, and Newman, 2008. Robustness of community structure in networks. Physical review E, 77(4), p.046119.

3) ICD codesTruncating the codes to three digits is akin to adding noise by losing information. Please justify your rationale behind truncating the codes. It'll be good if this issue could be addressed in the Discussion or one of the limitations of the work.

It is correct that we in principle and actually lose information when using level 3, but at the same time we would increase tremendously the dimensionality of the feature space if we went to level 4. In that sense, given the size of the data set where many level 4 codes will be unique, we likely would not be able to make a stable subsequent clustering and analysis. Using level 4 would mean that closely related level 4 codes (which would be same level 3 code) would be counted as unrelated. In fact, our attempt to use the much more fine-grained SNOMED-CT terminology confirmed that a data set needs to be very large for such a fine-grained vocabulary to be useful. The conclusion is likely very similar for ICD-10 level 4, in particular when taking the above remarks on the fuzziness of primary diabetes diagnoses into account. We have now elaborated further on the balance between information loss and coverage across patient features in the Discussion (new fifth paragraph).

4) Data limitations and temporal seriesFirst, there might be problems with the time-dependent comorbidities in a given cluster. There are multiple reasons: (1) the length of the observational window before and after considered in a given cluster is not talked about/defined. This can impact the enrichment of symptoms and hence all the downstream analysis; (2) length observational windows is bound to be different in different clusters, which may introduce the bias in computing comorbidities; (3) it is not clear from the analysis if there exist any relation between the observational window and cluster size.

These remarks are all relevant, but we think the reviewer has overlooked Figure 2—figure supplement 1 that describes the “Density of days in contact with SDCC for each cluster”. This additional characterization shows that we have long observational windows for a significant part of the cohort and that no cluster is considerably biased, or mixed time-wise. Given that the Steno Diabetes Center is a specialty clinic for severe diabetes cases where patients are admitted over long periods of time this is not surprising. This was part of the rationale for choosing this cohort.

Indeed, it is not clear how temporality is defined. The heatmap presented in Figure 2B does not help either. Specifically, what is the index date, relative to which pre and postcodes were identified? It is quite likely that the index date would vary for patients, therefore, how does that variability was accounted? Please explain this paragraph in more detail and if possible, then provide a schematic diagram representing temporality and how it was considered in this analysis. In addition, in the fourth paragraph of the subsection “Comorbidity pairs and patterns within symptom related clusters”, it appears the time window before and after is likely to bias the estimation of T2D and elevated blood glucose level. More clarity on this will be helpful.

We assume that the reviewer refers to Figure 4B and Figure 4 in general (and not Figure 2B). The figure is actually only used to describe the order of diagnosis in observed pairs. The actual time-difference analysis is not used elsewhere in the manuscript and we have therefore decided to remove the one sentence that mentions it. Similarly, the few stars that were inserted in the heatmap have been removed.

In any case, a better explanation is needed about the length of the observational window before and after used to compute comorbidities. The length of the observational window (before and after in a given cluster) is bound to confound the disease association. Was this length kept similar for all the clusters? If not, then it is very much possible that certain cultures are (just based on cluster size) are likely to be enriched in certain symptoms. Please provide an in-depth analysis on this.

This was per the comments above included already, or removed.

5) Additional dataPlease briefly explain the type of genetic data is included and at what level of omics.The candidate genetic and biochemical markers of glucose regulation could be of interest if replicated. However, could these have been discovered without the clustering e.g. using simpler regression methods?

Detailed information about the genetic data can be found in the “Genomic characterization” section under Materials and methods. We have inserted a reference to this section in the “Genomic characterization by SNP association of phenotypically determined clusters” section, where we first present results from this analysis to resolve any inaccuracies.

Further, we think the reviewer means glycemic dysregulation and not glucose regulation here. Per the remark below the genetic markers are analyzed without the clustering. We prefer not to go into the question of whether the biochemical markers could be discovered using a simple regression method. When one knows a pattern, or a signal, simpler “rules” or formulas can often be designed subsequently, but we do not think this is the case here.

Finally, we had by mistake left out references to the studies that generated the genetic data. These are now included (Charmet et al., 2018; Steinthorsdottir et al., 2014).

It is unclear how the clusters segregate in their genetic basis, which could then support the identification of disease subtypes. The question is not as much which genetic variants are associated with a given cluster, but if the effect estimations are statistically different between clusters, which would then favor the existence of subtypes. It is also unclear if the association signals are genome-wide significant in any case.

The genetic characterization of dysregulation analysis has been done independent of the clustering. We have realized that the section title “Genetic characterization of dysregulated patients and clusters” could indicate that the analysis is done cluster-wise, which is not the case due to low coverage of genetic data. We have changed the title accordingly to: “Genetic characterization of dysregulated patients*”.*

We agree that the segregation of genetic variants across clusters could provide additional support for disease subtypes. However, we think that such analysis should be performed in a cohort where the coverage of genetic data in the clusters are near complete. In our current setup, we believe that basing comparisons of effect estimates on varying sample sizes (that is number of patients with genetic data available in each cluster) is strongly under-powered and could lead to spurious results.

[Editors' note: further revisions were requested prior to acceptance, as described below.]

The manuscript has been improved but there is still some confusion about the significance of the resulting clusters, significance of the selected thresholds and the subsequent biological interpretation, as outlined below by the first reviewer. The final version should clarify the exploratory nature of the work and the limits of the statistical reliability, as pointed out by the second reviewer.Reviewer #1:The authors have done a perceptible effort trying to answer the major concerns initially raised, and I personally appreciate the clarifications and additional data provided.However, I am still not convinced that the main results are robust enough to support the overall message, in particular regarding the title: "Linking glycemic dysregulation.…to symptoms, comorbidities and genetics…".

In the paper we are exploring how symptoms, comorbidities and genetics are related to glycemic dysregulation, so we would argue that the title is appropriate. In our opinion, the word “linking” is not particularly loaded, or guaranteeing causal relationships and does not exaggerate the merits of the paper. Data linkage is not a term that in the literature is used to indicate causality. We have already in the text emphasized and described its explorative nature. Other titles could obviously be formulated, but we feel it is a bit hard to follow the reviewer’s concern here.

First, the issue of "circularity". I might have over-looked the original Discussion that mentioned that primary and secondary Diabetes diagnoses were excluded, or my initial comment might have been unclear, but I was referring to the expectation that clusters should obviously identify major known Diabetes types; in particular if dozens of diagnosis codes (some of them correlated) and extensive text vocabulary is analyzed. The point is not if the clusters identify what it is already known, but if they identify new etiology/subtypes.

We have not in the paper anywhere stated that the clusters represent diabetes subtypes in terms of etiology. They are descriptive in relation a full continuum diabetes population, and they are presented as a starting point for assessing to what extent the comorbidity/symptom-based subgrouping represents dissimilar etiologies, different or overlapping pathways at the molecular/trajectory levels, including genetic differences. Several recently published subgroupings have been presented as different etiological subtypes even if they presumably to a larger extent represent ethnic differences (when genetics is included in the feature space), or just physiologically similar subgroups that may not share genetics or other mechanistic features. We feel that we have been careful not to follow the same strategy. We present a comorbidity-based subgrouping that is linked to glycemic dysregulation. This subgrouping prompts precision medicine considerations as changes in disease co-occurrences link to differences in glycemic dysregulation and treatment planning. Precision medicine efforts should zoom in on these differences in outcomes, but as the paper is not a multi-omics effort across subgroups it would be unserious to present speculative, mechanistic models and claim them as different diabetes subtypes.

Of note, the following example is somewhat inaccurate: "… triple negative breast cancer patients share many features with ER-positive patients for example." Indeed TNBC and ER+ cases do not share cell-of-origin, prognosis, tissue-metastasis preference, neither therapeutic approaches.

This particular text in the response is referring to phenotypic similarity and, in that regards, TNBC and ER+ indeed share many similarities, both are breast cancers and therefore associated with many of the same types of symptoms to varying degrees. We do not at all disagree with the reviewer, we just highlight that subgroups of patients with diseases that are different also have many similarities, that is the whole point in subgrouping. In our case, we identify patient subgroups of e.g. cancer/diabetes co-occurrences that differ in terms of glycemic regulation. Please note that this text in not in the manuscript file, but only in the response document, and therefore we do not think we need to change our wording. This also links to the remarks about on similarities and dissimilarities in the context of precision medicine above.

My reasoning above is then linked to the title issue: the study of glycemic perturbation is presented in one section (“Glycemic dysregulation”) and remains unclear which clusters, symptoms, are robust regarding statistical differences for "glycemic perturbation" from the others. The thresholds of 0-2, 3-5 parameters, and then the clusters (21 with 50% patients 3 parameters…) appear to be arbitrary. The use of sentences with the following grammatical constructions may not help to understand the analyses: "distinct differentiation…dysregulation parameters".

The comparison between patients fulfilling 0-2 dysregulation parameters with patients fulfilling 3-5 is motivated by the hierarchical clustering of these groups presented in Figure 1—figure supplement 6. Patients in the 0-2 group are more similar to one another than those in the 3-5 group, and vice versa. Next, we chose to examine clusters where the majority of patients are in the “high” dysregulated group and therefore a cutoff of 50% is used. We have now rewritten the text to make this clearer.

Next, glycemic perturbation is linked to genetic risk in the last Results section, but as initially raised, it is unclear what means "top association signals"; are these significant genome-wide, are they statistically different between the clusters/symptoms? Are cluster-risk interactions significant? Honestly, this remains unclear to me and I hope I am not overlooking to any data.

We have rewritten the last part of the Results section to emphasize that the top associating signals do not reach genome-wide significance. We have not performed any comparisons of genetic risk or cluster-risk interactions between the clusters. While the suggestions from the reviewer are highly interesting, unfortunately, the low coverage of genetics for the clusters does not allow us to answer such questions. We have now also added a sentence to the genetics Materials and methods section with the definition of genome-wide significance in terms of p-value.